# DLCNet: Long-Range Convolution Need Data Dependency

## Abstract

In recent years, the Transformer architecture and self-attention mechanism have become the first choice for sequence modeling, but they encounter significant computational challenges when processing lengthy sequences. Long-range convolution has emerged as a promising alternative to self-attention, offering distinct advantages across various domains. However, current long-range convolution architectures still face several issues, such as excessive parameter usage and limited in-context learning capabilities. To tackle these challenges, we propose a Data-dependent Long-range Convolution Network (DLCNet) that introduces data dependency through three key modules: Layer-Wise Mapping, Rectify SideNet, and SWEAP Operator. DLCNet effectively facilitates in-context learning within a reasonable parameter scale. Extensive experiments have demonstrated that DLC-Net surpasses the state-of-the-art baselines in processing lengthy sequences, even when trained with short sequences.

## 1 Introduction

In the realm of Artificial Intelligence (AI) and Natural Language Processing (NLP), the emergence of Large Language Models (LLMs) has marked a transformative era (Press et al., 2022; Ouyang et al., 2022). These models, leveraging the influential Transformer architecture (Vaswani et al., 2017), undergo unsupervised training on extensive text corpora, showcasing impressive proficiency in understanding and generating human language. Their ability to grasp intricate patterns, contextual subtleties, and even emulate human-like conversational interactions has positioned LLMs at the forefront of cutting-edge research and practical applications (Chen et al., 2021; Brown et al., 2020; Touvron et al., 2023).

Despite the remarkable achievements of LLMs based on the Transformer architecture, the inherent self-attention mechanism in Transformer brings forth several significant challenges. One of these challenges is the quadratic time and memory complexity during computation, which results in a growing demand for computational resources and limitations when it comes to expanding the length of text they can handle. Additionally, these models tend to exhibit suboptimal parameter extrapolation characteristics, as demonstrated in recent research (Chen et al., 2023) when dealing with lengthier sequences.

In response to these challenges, a diverse array of alternative architectures have emerged, such as the Linear Transformer (Wang et al., 2020; Katharopoulos et al., 2020; Zhang et al., 2023), paralleled RNN (Stollenga et al., 2015; Peng et al., 2023), and State Space Model (Mehta et al., 2022; Gu et al., 2020; 2022). Among these approaches, the Long-Range Convolution has garnered substantial attention as a promising solution (Li et al., 2022; Qin et al., 2023; Poli et al., 2023). By harnessing global convolution to capture sequential information, the Long-Range Convolution achieves performance comparable to that of conventional Transformers while typically maintaining subquadratic time complexity, making it particularly well-suited for handling lengthier sequences.

The major drawback of Long-Range Convolution is that, compared to self-attention, its kernels remain unchanged across various input data, limiting its capability to deal with complex tasks. In this study, we propose a novel **D**ata-dependent **L**ong-range **C**onvolution **Net**work, denoted as **DLCNet**. It harnesses the efficiency of long-range convolutional computations while dynamically adapting convolution kernels to accommodate the input data.

First, to create a convolutional kernel with reasonable parameter count, we introduce a Layer-Wise Mapping technique to project a single adaptable decay rate onto multi-dimensional long convolution kernels, ensuring efficient mapping with minimal parameter expansion. The term 'Layer-Wise' underscores our approach of introducing initial features at different layers, each associated with distinct decay rates, thus enhancing performance.

Second, we propose the Rectify SideNet, which introduces data dependency into the convolution kernels. In conventional convolution processes, kernels remain unaltered across diverse input data. This lack of adaptability can impede advanced feature learning, potentially constraining the model's performance on complex tasks. To mitigate this limitation, we introduce the Rectify SideNet, proficient at extracting input data features and seamlessly integrating these features into data-independent convolution kernels, yielding highly effective outcomes.

Furthermore, we analyze the straightforward addition of input data to the convolutional kernel can introduce noise to the kernel itself. Consequently, we design a SWEAP Operator (Stable Weighted Exponential Average Pooling Operator). This operator serves to filter out the noise introduced by the input data, thereby ensuring the stability and reliability of the convolutional kernels.

Our main contributions can be summarized as follows:

- We introduce a Data-dependent Long Convolutional Network, or DLCNet for short. DLC-Net combines the computational efficiency of traditional convolutional networks on lengthy data sequences while addressing their limitation in adapting to different types of input data.

- We achieve this by incorporating three specialized modules into DLCNet. Specifically, we design Layer-Wise Mapping to obtain learnable convolutional kernels for different layers, and design Rectify SideNet and SWEAP Operator to efficiently leverage the input data to obtain updated data-dependent convolutional kernels.

- Extensive experiments demonstrate that DLCNet achieves mostly better or comparable performance on both self-supervised pretraining and downstream tasks. Moreover, DLCNet excels in generalization and extrapolation, even when applied to longer sequences not encountered during training.

## 2 PRELIMINARIES AND RELATED WORKS

### 2.1 TRANSFORMER AND ITS VARIANTS

The Transformer architecture (Vaswani et al., 2017), with its self-attention mechanism, has shown great proficiency in natural language processing (Brown et al., 2020; Touvron et al., 2023). However, a significant challenge arises due to quadratic time and memory complexities to the input sequence's length. These complexities pose significant challenges when dealing with lengthy sequences. Various approaches have been explored to address this issue, and we give a comparison of these methods in terms of data dependency and complexity in Table 1. One line of recent works involves revisiting the attention mechanism itself, e.g., Reformer (Kitaev et al., 2020), Performer (Choromanski et al., 2021), LS Attention (Zhong et al., 2019), AFT-full (Zhai et al., 2021). Another avenue of research has abandoned the attention mechanism entirely. For instance, RWKV (Peng et al., 2023) utilizes Recurrent Neural Networks (RNNs) to achieve linear time and memory complexity. S4 (Gu et al., 2022) and DSS (Gupta et al., 2022), on the other hand, simulate a fundamental State Space Model (SSM) to represent sequences. However, attention-based methods still face the problem of high complexity, RNN-based methods struggle with longer sequences, and SSM-based methods fail to achieve data dependency. Recent efforts have also explored the use of convolution to address these challenges, such as TNN (Qin et al., 2023) and Hyena (Poli et al., 2023). Nevertheless, these approaches also face challenges related to capturing data dependencies. In this work, we employ long-range convolution to efficiently model extended sequences and introduce data-dependent convolution kernels to enhance the model's expressive capabilities.

### 2.2 LONG-RANGE CONVOLUTION AND FFT-BASED ACCELERATION

Long-Range Convolution stands as a cutting-edge strategy harnessed in the realms of signal processing and deep learning. It comes to the fore when optimizing the computation of convolutions

Table 1: Comparison with the Transformer architecture and its variants. $T$ indicates sequence length and $d$ indicates hidden dimension.

| | Data Dependency | | Complexity | |
|---|---|---|---|---|
| | Token Mixing | Channel Mixing | Time | Memory |
| **Attention-based** | | | | |
| Transformer | ✔ | ✘ | $O(T^2d)$ | $O(T^2 + Td)$ |
| Reformer | ✔ | ✘ | $O(TlogTd)$ | $O(TlogT + Td)$ |
| Performer | ✔ | ✘ | $O(Td^2logd)$ | $O(Tdlogd + d^2logd)$ |
| LS Attention | ✔ | ✘ | $O(Trd)$ | $O(Lr + Ld + rd)$ |
| AFT-full | ✘ | ✘ | $O(T^2d)$ | $O(Td)$ |
| **RNN-based** | | | | |
| RWKV | ✔ | ✱ | $O(Td)$ | $O(Td)$ |
| **State Space Model-based** | | | | |
| S4 | ✔ | ✘ | $O(T(logT + logd)) + dlogT$ | $O(Td)$ |
| DSS | ✔ | ✘ | $O(TdlogT)$ | $O(Td)$ |
| **Convolution-based** | | | | |
| TNN | ✱ | ✱ | $O(TdlogT)$ | $O(Td$ |
| Hyena | ✱ | ✱ | $O(TdlogT)$ | $O(Td)$ |
| **DLCNet** | ✔ | ✱ | $\boldsymbol{O(TdlogT)}$ | $\boldsymbol{O(Td)}$ |

✔: fully dependency; ✘: no dependency; ✱: partial dependency (GLU-based dependency, etc.)

entailing extensive sequences becomes imperative. For an input sequence $\mathbf{X} \in \mathbb{R}^{l \times h}$ with length $l$ and hidden state dimension size $h$, the FFT accelerated convolution (Mathieu et al., 2014) on $\mathbf{X}$ can be formulated as:

$$\mathbf{O} = \text{FFT}(\mathbf{X}, \mathbf{K}) = \text{IFFT}\left(\text{FFT}(\mathbf{X}) \cdot \text{FFT}(\mathbf{K})\right), \tag{1}$$

where $\mathbf{K} \in \mathbb{R}^{l \times h}$ is the long-range convolution kernel and $\mathbf{O} \in \mathbb{R}^{l \times h}$ is the output. FFT represents the Fast Fourier Transform, and IFFT denotes the Inverse Fast Fourier Transform. The multiplication in the frequency domain is computationally efficient, particularly when dealing with long sequences, as it requires only element-wise multiplications.

If we focus on one channel of the whole hidden state dimension, for input sequence $\mathbf{x} \in \mathbb{R}^l$, the 1-$d$ convolution kernel $\mathbf{k} = [k_0, k_1, \cdots, k_{l-1}] \in \mathbb{R}^l$, a long-range convolution operation over the whole sequence can be expressed as:

$$\mathbf{K} \in \mathbb{R}^{l \times l}, \mathbf{K}_{ij} = \begin{cases} k_{i-j}, i \geq j \\ 0, i < j \end{cases}, \tag{2}$$

$$\mathbf{Kx} = \begin{bmatrix} k_0 & 0 & 0 & \cdots & 0 \\ k_1 & k_0 & 0 & \cdots & 0 \\ k_2 & k_1 & k_0 & \cdots & 0 \\ \vdots & \vdots & \vdots & \ddots & \vdots \\ k_{l-1} & k_{l-2} & k_{l-3} & \cdots & k_0 \end{bmatrix} \begin{bmatrix} x_0 \\ x_1 \\ x_2 \\ \vdots \\ x_{l-1} \end{bmatrix} = \begin{bmatrix} o_0 \\ o_1 \\ o_2 \\ \vdots \\ o_{l-1} \end{bmatrix}. \tag{3}$$

Clearly, in each dimension, long convolution, like models such as Transformer, adheres to the rules of casual decoding.

## 2.3 MIXING AND DATA DEPENDENCY

Without loss of generality, the operations performed on the input sequence $\mathbf{X}$ can be broadly divided into two categories as discussed in (Hè & Kabic, 2023).

**Token Mixing.** This involves left-multiplying $\mathbf{X}$ by a square matrix $\mathbf{A}$ of size $l \times l$. In essence, it's a process where information between different elements (tokens) within the matrix $\mathbf{X}$ is combined and adjusted. The result is represented as $\mathbf{O}$, where:

$$\mathbf{O} = \mathbf{AX}, \mathbf{A} \in \mathbb{R}^{l \times l}. \tag{4}$$

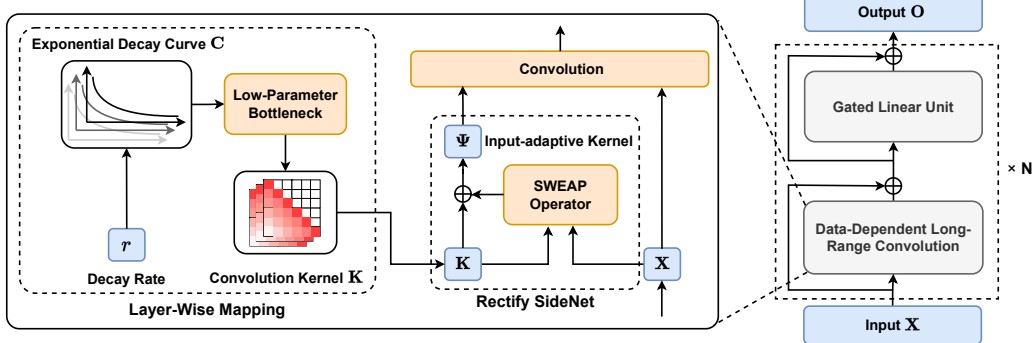

Figure 1: DLCNet contains three modules, Layer-Wise Mapping translates the decay rate $r$ into a convolution kernel $\mathbf{K}$; Rectify SetNet takes input $\mathbf{X}$ to dynamically transform the data-independent kernel $\mathbf{K}$ into an adaptive convolution kernel $\mathbf{\Psi}$; and SWEAP Operator eliminates input data noise to enhance the effectiveness of the convolution kernel $\mathbf{\Psi}$.

**Channel Mixing.** Here, we right-multiply $\mathbf{X}$ by a square matrix $\mathbf{A}$ of size $h \times h$. In this case, the aim is to adjust information from different aspects or channels present within the hidden states of $\mathbf{X}$. The result is represented as $\mathbf{O}$, where:

$$\mathbf{O} = \mathbf{X}\mathbf{A}, \mathbf{A} \in \mathbb{R}^{h \times h}. \tag{5}$$

Expanding upon this idea, we can delve into the crucial concept of data dependency. In the case where $\mathbf{A}$ remains unaltered regardless of variations in the input data, as in conventional convolution techniques (LeCun et al., 1995), we categorize both $\mathbf{A}$ and the entire operation as "**data-independent**". Conversely, when A exhibits variations in response to changes in the input data, as in Transformer (Vaswani et al., 2017), where $\mathbf{A}$ essentially represents the self-attention scores and is computed based on input data, we classify it as "**data-dependent**". In this context, the GLU (Dauphin et al., 2017) (Gated Linear Unit)-based dependency is a special case, which can be seen as left-multiplying $\mathbf{X}$ by a matrix that is diagonalized by itself:

$$\text{GLU}(\mathbf{X}) = \mathbf{X} \cdot \sigma(\mathbf{W_g} \cdot \mathbf{X}) = \begin{bmatrix} w_{g0} \cdot x_0 & 0 & 0 & \cdots & 0 \\ 0 & w_{g1} \cdot x_1 & 0 & \cdots & 0 \\ 0 & 0 & w_{g2} \cdot x_2 & \cdots & 0 \\ \vdots & \vdots & \vdots & \ddots & \vdots \\ 0 & 0 & 0 & \cdots & w_{g(l-1)} \cdot x_{l-1} \end{bmatrix} \begin{bmatrix} x_0 \\ x_1 \\ x_2 \\ \vdots \\ x_{l-1} \end{bmatrix}. \tag{6}$$

While this operation does not allow the mixing of information between different positions, it enables the entire operation's output to be controlled by itself. Therefore, we consider this to be a "**partially data-dependent**" method.

## 3 DATA-DEPENDENT LONG-RANGE CONVOLUTION

In this section, we present the proposed DLCNet. Similar to the architectural conventions of most Large Language Models (LLMs), DLCNet comprises two key components: token mixing and channel mixing. We accomplish token mixing through the proposed Data-Dependent Long-Range Convolution while employing the Gated Linear Unit (GLU) for channel mixing (Dauphin et al., 2017). Our principal innovation centers on the token mixing aspect, which encompasses three primary elements: **Layer-Wise Mapping**, **Rectify SideNet**, and **SWEAP Operator**.

**Layer-Wise Mapping** in Section 3.1 is a position-agnostic, low-parameter process capable of mapping a set of learnable decay parameters into multiple exponential decay convolution kernels. Its position-agnostic characteristic ensures excellent extrapolation capabilities. Furthermore, as the number of layers increases, the initialized decay rate decreases, allowing the model to preserve more macroscopic information in higher layers.

**Rectify SideNet** in Section 3.2 is responsible for generating kernel weights across different input data to rectify the original data-independent convolution kernels, thus introducing data dependency during the convolution process.

**SWEAP Operator** in Section 3.3, or the Stable Weighted Exponential Average Pooling operator, contributes to the reconstruction of kernels, transforming them from chaotic input data into a set of exponential-decay-like kernels, resulting in a significant improvement in our model's performance.

## 3.1 LAYER-WISE MAPPING

Traditionally, long convolutional kernels impose a significant parameter load, directly proportional to the product of their length $l$ and height $h$. This results in $l \times h$ learnable parameters in total. However, this approach becomes increasingly impractical as sequence length grows, causing models oversized and potentially hampering their overall performance (Li et al., 2022). Moreover, when the entire parameter matrix becomes learnable, each parameter becomes tightly coupled to its specific position within the sequence. Fixed parameters mean that when the model's input sequence length is extended, there won't be corresponding positional parameters for it, and its extrapolation ability will be weakened. Therefore, there arises a need for efficiently mapping a smaller subset of learnable parameters onto convolutional kernels. And the parameters should be decoupled from the sequence length, to ensure a promising extrapolation capability regardless the context (Qin et al., 2023).

Based on the aforementioned principles, we propose the Layer-Wise Mapping. Layer-Wise Mapping can generate convolution kernels with sub-linear parameter quantities, where stay in the best ability of extrapolation. Initially, there is typically a characteristic of information decay, meaning that information tends to diminish as we move further away from the current position. We often use exponential decay to describe this process. The exponential decay curve starts at a specific initial point, and as we move away from that point, the degree of information loss gradually increases. The decay speed is controlled by the decay rate. For a multi-layer model like Transformers, previous experiment results in Peng et al. (2023) show that the decay rate should decrease as the number of layers in the model increases. In other words, at lower layers, information typically decays rapidly, indicating that the model primarily focuses on local information. Conversely, at relatively higher layers, the model often requires a lower decay rate to attend to global information. Therefore, we consider designing different decay rates for layers from low to high. Specifically, we define the total number of layers in the model as $M$, and the layers are numbered from 0 to $M - 1$ , starting from the lowest layer. With a base decay rate of $r_{init}$, to shift down the decay speed as the number of layers increases, the decay rate $r_m$ in layer $m$ can be expressed as follows:

$$r_m = \frac{r_{init} + \tau(m+1)}{M} \in (0,1), \tau \in (0,1). \tag{7}$$

Here $\tau$ is a fixed layer-wise penalty factor used to control the decay of the layer number on the decay effect. As the number of layers increases, the value of $r_m$ becomes larger, leading to a lower decay speed. With the decay rate, we further extend it to a decay curve. We follow the settings in most recent works to introduce an exponential decay curve which has been experimentally proven to be the most effective among various decay methods (Qin et al., 2023). This process is expressed as:

$$\mathbf{C}^m = [(r_m)^0, (r_m)^1, \cdots, (r_m)^{l-1}] \in \mathbb{R}^{l \times 1}. \tag{8}$$

Now $\mathbf{C}^m$ can be considered as the exponential decay curve in layer $m$ constructed on the basis of $r_m$, and $(r_m)^p$ denotes the value at the $p$-th position equals to the value of $p$-th power of $r_m$. As the position increases, the decay value gradually diminishes towards zero. However, in our earlier derivation, there are no other trainable parameters aside from $r_m$. This implies that the curve of $\mathbf{C}^m$ remains relatively unchanged during training, considerably elevating the training complexity. Therefore, additional steps are required to render $\mathbf{C}^m$ learnable.

To do that, we extend this non-trainable curve $\mathbf{C}^m$ to all hidden state channels and introduce learnability. Additionally, we aim to prevent overfitting, as employing a significant number of learnable parameters could lead to increased complexity, high computational cost, and the risk of overfitting. Therefore, we employ a concise solution: a MLP network with low-parameter bottleneck.

First we expand $\mathbf{C}^m$ along the last dimension to a bottleneck dimension $d$ that is smaller than the hidden dimension $h$:

$$\mathbf{C}_0^m = \mathbf{C}^m \mathbf{W}_0, \mathbf{C}^m \in \mathbb{R}^{l \times 1}, \mathbf{W}_0 \in \mathbb{R}^{1 \times d}, \mathbf{C}_0^m \in \mathbb{R}^{d \times d}. \tag{9}$$

Here $h$ is the dimension of the hidden state. Then $\mathbf{C}_0^m$ will go through $L$ linear layers, for the $i$-th layer, this process can be expressed as:

$$\mathbf{C}_i^m = \text{SiLU}(\mathbf{C}_{i-1}^m \mathbf{W}_i), \mathbf{C}_i^m \in \mathbb{R}^{l \times d}, \mathbf{W}_i \in \mathbb{R}^{d \times d}. \tag{10}$$

As shown in Section 2.2, all the operations mentioned above are channel mixing. Therefore, the number of parameters in $\mathbf{W}_i$ is independent of the context length. As a result, the number of parameters does not increase with the growth of sequence length. Finally we get the $\mathbf{C}_L^m \in \mathbb{R}^{l \times h}$:

$$\mathbf{C}_L^m = \mathbf{C}_{L-1}^m \mathbf{W}_L, \mathbf{W}_L \in \mathbb{R}^{d \times h}. \tag{11}$$

As we mentioned in Section 2.2, in the FFT accelerated convolution, we pass $\mathbf{X}, \mathbf{K} \in \mathbb{R}^{l \times h}$ to the operation $\mathrm{FFT}(\mathbf{X}, \mathbf{K})$ and the output is $\mathbf{O} \in \mathbb{R}^{l \times h}$. Apparently, $\mathbf{C}_L^m$ is the $\mathbf{K}$ we need in layer $m$.

It's important to note that while we represent the convolution process on each dimension as a $\mathbf{K}$ matrix of size $l \times l$ left-multiplied on our input sequence $\mathbf{X}$, in the actual implementation, we perform a fast Fourier transform (FFT) on both the K and X matrices, which are both of size l*h. After multiplication, we then perform the inverse transform.

## 3.2 RECTIFY SIDENET

Another notable aspect of DLCNet is the proposed Rectify SideNet. As Equation 1 in Section 2.2, in traditional long-range convolutions, regardless of the layer number $m$, the convolution process can be simply expressed as: $\mathbf{O} = \mathbf{FFT}(\mathbf{X}, \mathbf{K}), \mathbf{X} \in \mathbb{R}^{l \times h}, \mathbf{K} \in \mathbb{R}^{l \times h}$. The kernel $\mathbf{K}$ is independent to input data and remains unchanged. This inflexibility potentially constrains the model's performance across complex tasks. To address this limitation, we introduce data dependence into $\mathbf{K}$, enabling it to adapt to different input data.

We first consider expanding the width of the convolution analogous to the multi-head attention mechanism in Transformers. In other words, we use $h$ convolutional kernels at each layer per channel. In this scenario, our convolutional kernel $\mathbf{K}$ can be represented as a combination of $h$ sub-convolutional kernels for each channel in hidden state dimension:

$$\mathbf{K} = [\mathbf{k}_0, \mathbf{k}_1, \cdots, \mathbf{k}_{h-1}], \mathbf{k}_i \in \mathbb{R}^{l \times 1}.$$

To adaptively tailor $\mathbf{K}$ to input $\mathbf{X}$, we transform $\mathbf{X}$ to a data-dependent matrix $\mathbf{D}$:

$$\mathbf{D} = \mathrm{Sigmoid}(\mathbf{X}\mathbf{W}_D), \tag{12}$$

where $\mathbf{W}_D \in \mathbb{R}^{h \times h}$ is a trainable matrix. We use $\mathbf{D}$ to assign adaptive weights among different sub-convolutional kernels $\mathbf{k}_i$. Specifically, we reformulate $\mathbf{D}$ as a series of vectors as $\mathbf{D} = [\mathbf{d}_0, \mathbf{d}_1, \cdots, \mathbf{d}_{h-1}], \mathbf{d}_i \in \mathbb{R}^{l \times 1}$, and use $\mathbf{K}$ and $\mathbf{D}$ as input to obtain a adaptive kernel $\boldsymbol{\Psi}$ for each sub-kernels $\mathbf{k}_i$ in $\mathbf{K}$:

$$\boldsymbol{\Psi} = [\boldsymbol{\psi}_0, \boldsymbol{\psi}_1, \cdots, \boldsymbol{\psi}_{h-1}] = [\mathbf{k}_0 \odot \mathbf{d}_0, \mathbf{k}_1 \odot \mathbf{d}_1, \cdots, \mathbf{k}_{h-1} \odot \mathbf{d}_{h-1}], \tag{13}$$

where $\boldsymbol{\Psi}$ has the same size of $\mathbf{K}$ and $\odot$ denotes element-wise product. We can simplify the above formula as:

$$\boldsymbol{\Psi} = \mathbf{K} \odot \mathbf{D}. \tag{14}$$

Finally, we employ the FFT-accelerated convolution with the input-adaptive kernel $\boldsymbol{\Psi}$:

$$\mathbf{O} = \mathrm{FFT}(\mathrm{SiLU}(\mathbf{X}\mathbf{W}_V), \boldsymbol{\Psi}), \tag{15}$$

where $\mathbf{W}_V \in \mathbb{R}^{h \times h}$ is a trainable matrix and $\mathbf{O} \in \mathbb{R}^{l \times h}$ is the convolution output.

## 3.3 SWEAP OPERATOR

We aspire to maintain the exponential decay form of $\mathbf{D}$ during the rectification process to ensure that the output remains the exponential decay characteristics. Nevertheless, since $\mathbf{K}$ exhibits a tendency to decay exponentially, we postulate that $\mathbf{D}$ also needs to exhibit this same tendency otherwise noise will be introduced. As shown in Figure 2, we select 16 channels in $\mathbf{D}$ for visualization. The horizontal axis is the length of the input sequence and the vertical axis is the numerical value of the convolution kernel. For a convolution kernel $\mathbf{K}$ composed of exponential decay curves, the absence of an exponential decay trend in $\mathbf{D}$ can be considered a form of noise. Multiplying $\mathbf{K}$ by $\mathbf{D}$ will cause $\mathbf{K}$ to lose some of its exponential decay characteristics, and we find such loss significantly impact the model's learning capacity (in Section 4.3). Thus, our current inquiry is: Can we also transform $\mathbf{D}$ into a low-noise exponential decay form?

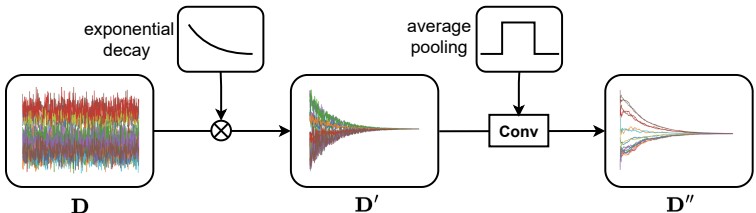

Figure 2: The SWEAP Operator. For the input $\mathbf{D}$ which contains noise, first the $\mathbf{D}'$ with exponential decay form is obtained by multiplying it using the exponential decay kernel $\mathbf{K}$, and then the filtered $\mathbf{D}''$ is obtained by using average pooling.

To elaborate on how to transform $\mathbf{D}$ (defined in Equation 12) into an exponential decay form and get the final data-dependent kernel $\mathbf{\Psi}$ in Equation 15, we design a Stable Weighted Exponential Averaging Processing (SWEAP) Operator. This operator optimizes the process of obtaining $\mathbf{\Psi}$ in Equation 14, and mainly consists of two parts: adding exponential decay factors and performing sliding window exponential decay (shown in Figure 2). Firstly, to ensure that the curve exhibits an exponential decay form overall, we multiply $\mathbf{D}$ by the exponential decay kernel $\mathbf{K}$:

$$\mathbf{D}' = \mathbf{K} \odot \mathbf{D}. \tag{16}$$

Now $\mathbf{D}'$ is in the form of exponential decay. However, $\mathbf{D}'$ still contains tons of noise. To filter out these noises, we draw on the idea of sliding window averaging and design a Stable Weighted Exponential Averaging operator. This operator uses a sliding convolution kernel with a window size $w$ to perform convolution over the entire curve with a learnable window kernel:

$$\mathbf{D}'' = \text{SWEAP}(\mathbf{D}', \mathbf{K_{window}}) = \frac{\text{FFT}(\mathbf{D}', \mathbf{K_{window}})}{\text{FFT}(\mathbf{E}, \mathbf{K_{window}})}. \tag{17}$$

$\mathbf{K_{window}} \in \mathbb{R}^w$ is a sliding window kernel of size $w$ smaller than $l$, containing a set of learnable parameters $[p_0, p_1, \cdots, p_{w-1}]$. In practice, we implement $\mathbf{K_{window}}$ with zero padding as $[p_0, p_1, \cdots, p_{w-1}, 0, \cdots, 0] \in \mathbb{R}^l$, and employ a denominator $\text{FFT}(\mathbf{E}, \mathbf{K_{window}})$ serves a regularization to eliminate the influence of padding, where $\mathbf{E} = [1, 1, \cdots, 1] \in \mathbb{R}^l$. We add the stable and exponential-decay style kernel $\mathbf{D}''$ to the original kernel $\mathbf{K}$ to obtain the input-adaptive kernel $\mathbf{\Psi}$ defined in Equation 14, and do an additional normalization on $\mathbf{D}''$ for the purpose of stabilizing the training process. Formulaically,

$$\mathbf{\Psi} = \mathbf{K} + \frac{\mathbf{D}''}{\sqrt{\sum_{i=1}^{l} \sum_{j=1}^{h} |\mathbf{D}''_{ij}|^2}}. \tag{18}$$

We then use this $\mathbf{\Psi}$ as a kernel of Equation 15 to finally get the output of the convolution operation.

## 4 EXPERIMENT

In this section, we present the following research questions as a guide to conduct empirical investigations. (**RQ1**) Can the DLCNet achieve state-of-the-art pretraining performance on large-scale corpus? (**RQ2**) Can the DLCNet achieve state-of-the-art performance across various downstream tasks? (**RQ3**) What is the contribution of each individual DLCNet module to the final performance? (**RQ4**) Can the DLCNet generalize and extrapolate for inputs longer than the training sequences?

### 4.1 PRETRAINING PERFORMANCE (RQ1)

We validate the pretraining performance on the commonly used WikiText-103 dataset (Merity, 2016). We use the pretraining perplexity (PPL) as the evaluation metric, and report the results after training 50K steps on the datase. The detailed experimental settings are illustrated in Appendix A.2. And the following groups of methods are selected as compared baselines, including (1) *attention-based*, GPT2, GPT-Neo, Performer (Choromanski et al., 2021), Reformer (Kitaev et al., 2020), Linear Attention (Zhong et al., 2019), AFT-Conv (Zhai et al., 2021); (2) *rnn-based*, RWKV (Peng et al., 2023); (3) *ssm-based*, H3(Fu et al., 2023), S4 (Gu et al., 2022); (4) *cnn-based*, Hyena (Poli et al., 2023). Table 2 reports the results on WikiText-103. It is evident that DLCNet achieves performance on par with other Linear Attention architectures while maintaining lower time and space complexity. This standard will be consistently applied in our subsequent evaluations.

Table 2: Pretraining PPL on WikiText-103. The best and second best for each metric are in **bold** and underlined format, respectively.

|  | attention-based | | | | | | ssm-based | | cnn-based | |
|---|---|---|---|---|---|---|---|---|---|---|
|  | GPT2 | GPT-Neo | Performer | Reformer | Linear Attention | AFT-conv | H3 | S4 | Hyena | DLCNet |
| Params | 137M | 125M | 125M | 125M | 125M | 125M | 125M | 249M | 125M | 128M |
| PPL ↓ | 29.9 | 26.3 | 26.8 | 25.6 | 25.6 | 28.2 | 23.7 | 21.0 | **18.6** | 20.8 |

## 4.2 Performance on Downstream Tasks (RQ2)

We proceed to evaluate the performance of DLCNet on downstream tasks. In this experiment, we select the following methods as baselines: RWKV-4 (Peng et al., 2023), GPT-2 (Radford et al., 2019), GPT-Neo (Black et al., 2021), Pythia (Biderman et al., 2023), Hyena Poli et al. (2023). For the model trained on the WikiText-103 dataset, we test its in-context learning ability on the SuperGLUE benchmark (Wang et al., 2019). For the model trained on the Pile, we employ several natural language understanding datasets as the downstream task: Lambada (Storks et al., 2020), PIQA (Bisk et al., 2019), StoryCloze (Roemmele et al., 2011), COPA (Mostafazadeh et al., 2017), Winogrande (ai2, 2019), ARC (Yadav et al., 2019), SciQ (Welbl et al., 2017), OpenBookQA (Mihaylov et al., 2018). In the SuperGLUE benchmark, we compared our language understanding capabilities with Hyena.

Table 3: Performance on SuperGLUE benchmark.

|  | Params (M) | WSC (acc↑) | WIC (acc↑) | RTE (acc↑) | CB (acc↑) | MultiRC (acc↑) | ReCoRD (acc↑) | BoolQ (acc↑) | Average (acc↑) |
|---|---|---|---|---|---|---|---|---|---|
| Hyena | 125 | 21.2 | 50.5 | **46.6** | 39.3 | **1.1** | 59.4 | 51.8 | 38.6 |
| DLCNet | 128 | **22.3** | **51.2** | 45.3 | **41.5** | 0.0 | **59.5** | **55.6** | **39.4** |

It can be observed in Table 3 that, although DLCNet's perplexity during pre-training was worse than that of Hyena, DLCNet outperforms Hyena in downstream tasks across various scenarios with the same parameter count. Results in Table 4 on other datasets also demonstrate that DLCNet, while maintaining superior time and space complexity compared to traditional transformer architectures, can achieve performance on par with transformers.

Table 4: Performance on downstream natural language understanding tasks. The best and second best for each metric are in **bold** and underlined format, respectively.

|  | Params (M) | Lambada (ppl↓) | PIQA (acc↑) | StoryCloze (acc↑) | COPA (acc↑) | Winogrande (acc↑) | ARC-e (acc_norm↑) | ARC-c (acc↑) | SciQ (acc↑) | OBQA (acc_nrom↑) |
|---|---|---|---|---|---|---|---|---|---|---|
| RWKV-4 | 169 | 29.33 | **65.07** | 58.79 | **66.00** | 50.83 | **47.47** | **24.15** | 77.50 | **29.60** |
| GPT-2 | 137 | 40.11 | 62.9 | 51.6 | 64.00 | 51.62 | 39.48 | 22.70 | - | 16.40 |
| GPT-Neo | 125 | 30.27 | 63.06 | 58.26 | 64.00 | 50.43 | 43.73 | 23.12 | 76.50 | 26.20 |
| Pythia | 160 | **24.38** | 62.68 | 58.47 | 64.00 | **52.01** | 45.12 | 23.81 | 76.60 | 29.20 |
| DLCNet | 128 | 28.59 | 63.61 | **58.92** | **66.00** | 51.78 | 45.41 | 23.89 | **78.10** | **29.60** |

## 4.3 Ablation Experiment (RQ3)

We conduct a series of ablation experiments to analyze each component of DLCNet and explore their respective contributions to the final performance.

***Analysis of Layer-Wise Mapping.*** Firslty, we analyze the performance contributions of Layerwise Mapping module proposed in Section 3.1. Here we consider three different variants, namely: (1) **Dim**, different dimensions in the bottleneck network,with Low representing dimension $d = h/8$, and High representing $d = h$, where $d$ and $h$ are bottleneck and hidden dimension, respectively; (2) **Layer-Wise**, whether it has a layer-wise decay rate, if not, all layers share the same decay rate $r_{init}$; (3) **Mapping**, whether mapping the exponential decay in one dimension to the entire hidden state, if not, it is straightforward to make all hidden states share the same kernel by repetition. We evaluate these variants with the pretraining loss after 50000 steps under the same setting in Section 4.1. The

results are in the first group in Table 5. Clearly, the parameterized curves require mapping, and the low-parameter mapping along with varying decay rates per layer to some extent reduces the complexity of curve mapping and enhances the model's capabilities.

***Analysis of Rectify SideNet and SWEAP Operator.*** Next we investigate the effects of removing the Rectify SideNet (Section 3.2) and SWEAP Operator (Section 3.3) through the following variants: (1) **w/o SideNet**, removing the Rectify SideNet to disable the ability obtaining data-dependent convolutional kernels; (2) **w/o SWEAP**, removing the SWEAP Operator to understand its role to the reconstruction of kernels from chaotic input data; (3) **w/o SideNet & SWEAP**, removing the Rectify SideNet and SWEAP Operator together. We evaluate these variants with the pretraining loss after 50000 steps under the same setting in Section 4.1. The results are in the second group in Table 5, which clearly demonstrate that in the absence of the SWEAP Operator, the Rectify SideNet exhibits limited improvement. This aligns with our speculation that unprocessed data may introduce noise to the convolutional kernels. We give the results of visualizing the role of the SWEAP Operator in Appendix A.3.

Table 5: Results of ablation experiments.

| Dim | Layer-Wise | Mapping | PPL ↓ |
|---|---|---|---|
| Low | ✔ | ✔ | **20.8** |
| High | ✔ | ✔ | 21.7 |
| Low | ✘ | ✔ | 21.5 |
| High | ✘ | ✔ | 22.3 |
| Low | ✔ | ✘ | 22.5 |
| High | ✘ | ✘ | 23.1 |
| DLCNet | | | **20.8** |
| w/o SideNet | | | 21.1 |
| w/o SWEAP | | | 21.0 |
| w/o SideNet & SWEAP | | | 21.9 |

### 4.4 LONG TEXT LENGTH EXTRAPOLATION TEST (RQ4)

We conduct experiments on the Wikitext-103 to validate the extrapolation ability of the model. Specifically, we use input sequences of length 512 during training, and then randomly select 1000 sequences of length ranging from 512 to 10240 during testing. We employ PPL as a criterion to measure the extrapolation performance in an autoregressive manner. Figure 3 presents the results, with some baseline results taken from Qin et al. (2023). It is demonstrated that DLCNet exhibits excellent extrapolation capabilities, as its text perplexity remains consistent regardless of the increase in input text length, and it has a consistent performance advantage over other methods, thereby validating the effectiveness of our model design.

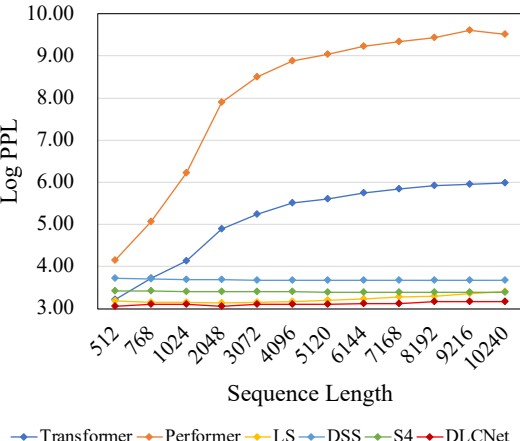

Figure 3: Extrapolation evaluation.

## 5 CONCLUSION

In this paper, we propose a data-dependent long-range convolution method for efficiently processing long sequences. Our method realizes the advantage of low time and memory complexity with the benefit of convolution operations. We address the problem that traditional convolution is data-independent by proposing three modules, namely Layer-Wise Mapping, Rectify SideNet, and SWEAP Operator, to effectively incorporate the information from the input data. Experimental results demonstrate that DLCNet exhibits excellent performance on both the self-supervised pretraining task and the downstream natural language understanding task. Notably, DLCNet has excellent extrapolation and generalization characteristics, and is able to maintain good performance on inputs that are longer than those encountered during training, which proves the superiority of our approach in dealing with long sequences.

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

# A  APPENDIX

## A.1  FFT ACCELERATED CONVOLUTION

In each channel of the Long-Range Convolution, we denotes that the following operation is done:

$$O = TX, T \in \mathbb{R}^{l \times l}, X \in \mathbb{R}^{l \times h} \tag{19}$$

We can optimize this matrix multiplication through FFT.

$$
\begin{bmatrix} y_0 \\ y_1 \\ y_2 \\ \vdots \\ y_{n-1} \end{bmatrix}
=
\begin{bmatrix}
t_0 & 0 & 0 & \cdots & 0 \\
t_1 & t_0 & 0 & \cdots & 0 \\
t_2 & t_1 & t_0 & \cdots & 0 \\
\vdots & \vdots & \vdots & \ddots & \vdots \\
t_{n-1} & t_{n-2} & t_{n-3} & \vdots & t_0
\end{bmatrix}
\begin{bmatrix} x_0 \\ x_1 \\ x_2 \\ \vdots \\ x_{n-1} \end{bmatrix}
\tag{20}
$$

Firstly, we observe that matrix $T$ is a discrete matrix. We can expand the matrix $T$ into a cyclic matrix, and at the same time fill in the $X$ matrix with zeros, so that the value of $Y$ does not change.

Afterwards, zero fill the vectors $Y$ and $X$ so that their lengths reach $m$ ($m$ is the smallest power of 2 greater than or equal to $n$). The length of the filled vector $T$ is $m$, and the length of vector $X$ is $m$.

Perform Fast Fourier Transform (FFT) on the zeroed vectors $T$ and $X$, respectively, to obtain their frequency domain representations. Record as $T_{hat}$ and $X_{hay}$.

$$T_{hat}[k] = \sum_{n=0}^{N-1} t[n] \cdot \exp\left(-\frac{2\pi i k n}{N}\right) k \in [0, m-1] \tag{21}$$

$$X_{hat}[k] = \sum_{n=0}^{N-1} x[n] \cdot \exp\left(-\frac{2\pi i k n}{N}\right) k \in [0, m-1] \tag{22}$$

Multiply these two vectors point by point to obtain the frequency domain representation of cyclic convolution. Mark as $Y_{hat} = T_{hat} * X_{hat}$.

Then perform inverse Fourier transform (IFFT) on the obtained vector to obtain the first $n$ bits of the vector, which is the matrix $Y$ we need.

$$Y[k] = \left(\sum_{l=0}^{m-1} Y[l] \cdot e^{i2\pi \frac{lk}{m}}\right)/m, \quad k \in [0, n-1] \tag{23}$$

In this way, we optimized the complexity from $O(n^2)$ to $O(n log n)$.

If we assign

$$
T =
\begin{bmatrix}
t_0 & 0 & 0 & \cdots & 0 \\
t_1 & t_0 & 0 & \cdots & 0 \\
t_2 & t_1 & t_0 & \cdots & 0 \\
\vdots & \vdots & \vdots & \ddots & \vdots \\
t_{n-1} & t_{n-2} & t_{n-3} & \cdots & t_0
\end{bmatrix}
, X =
\begin{bmatrix} x_0 \\ x_1 \\ x_2 \\ \vdots \\ x_{n-1} \end{bmatrix}
\tag{24}
$$

this operation can be rewritten as: which also satisfies the simplified form of $Self - AttentionMechanism$. Also, it is clear to see that the above operations can be accelerated using FFT to achieve a complexity of $O(n \log n)$.

It's important to note that we have omitted the hidden states here. In NLP tasks, each token is typically represented as a hidden state vector. In actual FFT computations, the formula is usually expressed as follows: We use vector cross product here:

$$\mathbf{r} = (r_0, r_1, r_2, \ldots, r_{n-1}) \in \mathbb{R}^h$$

$$\mathbf{x} = (x_0, x_1, x_2, \ldots, x_{n-1}) \in \mathbb{R}^h$$

$$\mathbf{o} = \mathbf{r} \odot \mathbf{x} = (r_o \times x_0, r_1 \times x_1, \cdots, r_{n-1} \times x_{n-1}) \in \mathbb{R}^h \quad (25)$$

We can say that $\mathbf{TX}$ is the operation of FFT convolution within one dimension of the hidden state. The entire FFT convolution can be understood as performing every $\mathbf{TX}$ operations within $h$ channels:

$$FFT(\mathbf{X}, \mathbf{K}) \to (\mathbf{TX}) \times htimes \to [\mathbf{T}_0 \mathbf{X}_0, \mathbf{T}_1 \mathbf{X}_1, \cdots, \mathbf{T}_{h-1} \mathbf{X}_{h-1}].$$

## A.2 EXPERIMENTAL SETTINGS

We conducted pre-training on a server equipped with V100 × 4 GPUs. Following the same benchmarks as other test models, we configured the model with 12 layers, a hidden layer dimension of 512, and a text length of 512. Detailed parameter settings can be found in the following table:

| DLCNet HyperParameter Settings for Pile in 162M | |
|---|---|
| | ddp |
| Precision | fp16 |
| Optimizer | Adam |
| Optimizer momentum | 0.9 0.98 |
| Peak learning rate | 6e-4 |
| Warmup learning rate init | 1e-5 |
| Final learning rate | 1e-5 |
| Weight decay | 0.1 |
| Learning rate schedule | Linear |
| Warmup schedule | cosine |

We initially conducted training for 50,000 steps on the Wikitext-103 dataset. Subsequently, we evaluated the perplexity of the generated text on the test set. Following that, we retrained the model on The Pile dataset, and after training on 300 billion tokens, we conducted zero-shot testing of its in-context learning ability on the SuperGLUE dataset.

## A.3 FURTHER ANALYSIS OF SWEAP OPERATOR

We analyze $\mathbf{D}$ (defined in Equation 12) that has not been processed by the SWEAP Operator in Figure 4 and $\mathbf{D}''$ (defined in Equation 17) that is processed by the SWEAP Operator with different window sizes in Figure 5. We select 16 channels for visualization. The horizontal axis is the length of the input sequence and the vertical axis is the numerical value of the convolution kernel. We observe that the information obtained without SWEAP Operator processing could not exhibit an exponential decay trend, but this trend appears after SWEAP Operator processing. And as the window increases, the curve gradually shows an exponential decay and becomes smoother.

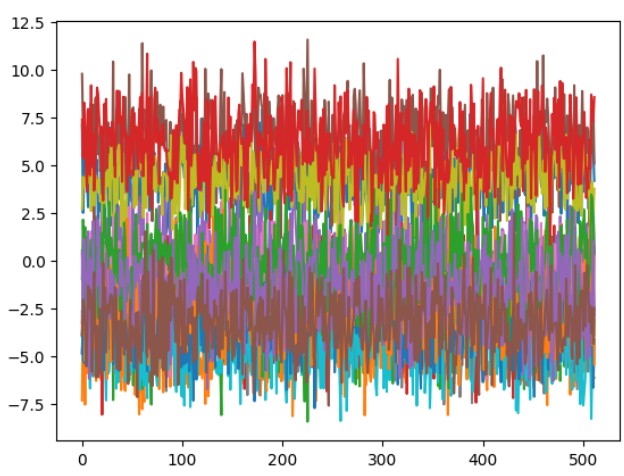

Figure 4: Visualization of $\mathbf{D}$.

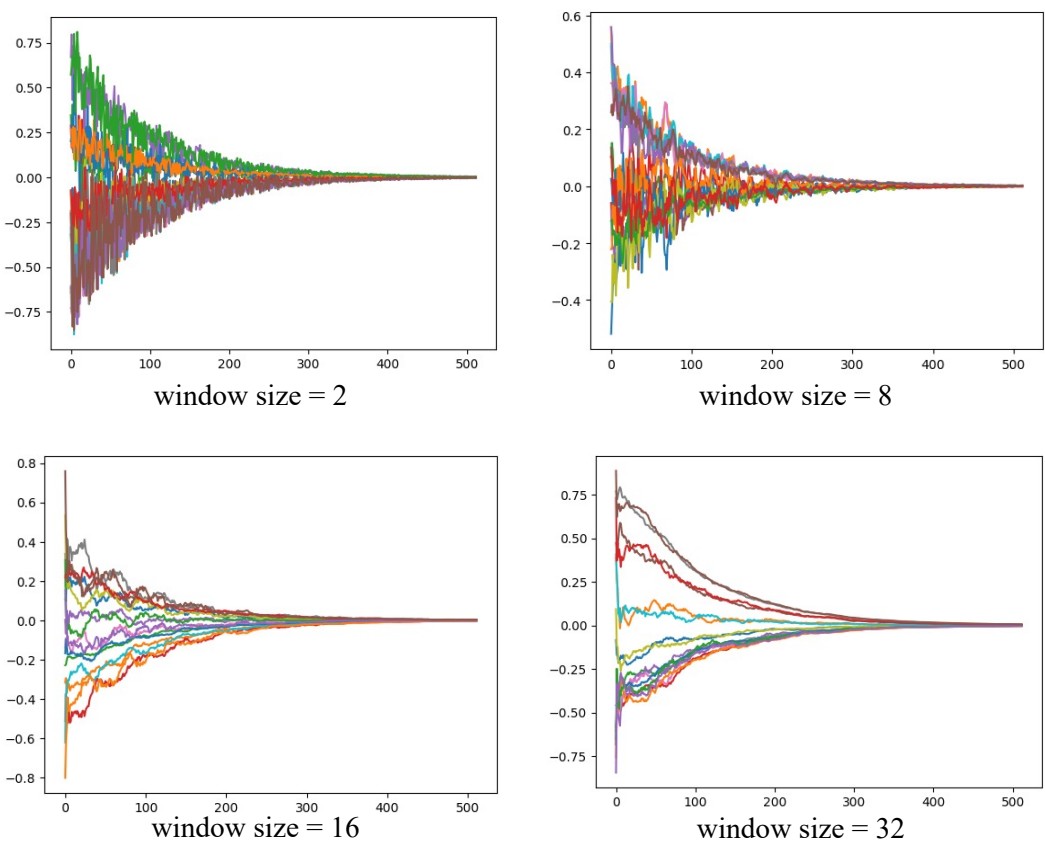

Figure 5: Visualization of $\mathbf{D}''$ with different window size.

