# OpenReview forum: "DLCNet: Enabling Long-Range Convolution with Data Dependency"
_ICLR.cc/2024/Conference — Submitted to ICLR 2024_

### Official Review · Reviewer_zAJh · 2023-10-31

**Soundness:** 2 fair
**Presentation:** 2 fair
**Contribution:** 2 fair
**Rating:** 3
**Confidence:** 3

**Summary:**

The authors study the use of long-range convolutions in large language models to model long range dependencies. They propose three new mechanisms for improving these models: “layer-wise mapping” to translate a decay rate into a convolutional kernel, “rectify sidenet” to condition the convolutional kernel on the input data, and a “SWEAP operator” to denoise the convolutional kernel. They study their proposed architecture on pre-training and downstream language tasks and compare to a range of alternative sequence learning architectures.

**Strengths:**

The authors compare their method to a wide range of alternative architectures including 6 different attention-based models, two state-space models and an alternative convolution based architecture.

**Weaknesses:**

I found this paper challenging to read and understand the proposed methods. The text through page 4 does little to explain the three mechanisms that have been proposed and I had to re-read a number of times to understand how the three proposed changes align with the descriptions of existing long-range convolutional architectures. One portion of the method that I am still unclear about is the decay used in the “layer-wise mapping”. It’s not clear to me if this is from existing long-range convolutional architectures or if this is a new idea the authors are introducing.

I also found the experimental results to be not very compelling. First, I think the authors should include the number of math operations that are done by each model and not just their parameter counts (e.g., in Table 2). Convolutional architecture often reuse parameters, which can lead to having higher quality per parameter but not necessarily per FLOP. This seems to be potentially true for the lower-parameter bottleneck in the proposed method as well. I’m particularly interested in these numbers for comparison with Hyena, which is quite close to the proposed architecture in quality and parameter count across pre-training and downstream tasks.

**Questions:**

GLU and SwiGLU architectures are gaining popularity in LLMs but I don’t believe any of the baselines in Section 4 use them? I’d be curious to understand the impact of using GLU on the quality and efficiency of the proposed architecture.

---

### Official Review · Reviewer_2j7h · 2023-11-01

**Soundness:** 3 good
**Presentation:** 2 fair
**Contribution:** 3 good
**Rating:** 6
**Confidence:** 2

**Summary:**

While attention-based transformer models show superior performance to convolutional neural networks on processing lengthy sequences, they require quadratic memory/time complexities with respect to the input length. To address this, some recent movements introduce adaptations of convolution that can capture global information of long sequences, denoted as long-range convolution network. Along this line, this paper proposes a new form of long-range convolution network that introduces the notion of “data-dependency” to the convolution kernel, which is originally the same across different inputs. To do this, the authors introduce three new operations to the architecture: 1) Layer-Wise Mapping technique that projects a single learnable value to several kernels to keep the network low-parameter 2) Rectify SideNet that extracts input data features and integrates it to the kernel 3) SWEAP Operator that filters out the noise introduced by input data integration. Experiments on NLP based tasks show that they can outbeat transformer-based models and achieve new SOTA results.

**Strengths:**

- The idea of making convolution kernel data-dependent is very interesting and poses great novelty. And each component added to the network is well-reasoned.
- The explanation of each component is detailed and makes appropriate reference to prior works to help understand previous findings and the consensus made upon prior works.
- The experiment is designed to address the necessary questions, including performance on both pretraining and down stream tasks, ablation study on the proposed method, the method’s efficacy on extremely long sequences.

**Weaknesses:**

- Since the use of long range convolution is rather recent, the authors should pay extra efforts to introduce this line of work in detail, especially in the related works section.
Also, a method-wise comparison against prior works such as Hyena and TNN should be provided to help the readers understand and make fair comparisons.
- Performance-wise, the results on downstream NLP tasks show that DLCNet is mostly second to other baselines such as RWKV-4, and in the cases where DLCNet outwins the margin is comparably small than in the opposite case.

**Questions:**

- In table 3, the authors make comparison only against Hyena. Is there a specific reason for doing so? If not, an extension of this table would be needed. (For example, Hyena includes GPTNeo and RWKV in the same experiment in their work.)
- Since this work has its value in incorporating data-dependency to the convolution network, it would be interesting to see how the filters actually change according to the input, and essentially “how” the kernel incorporates or utilizes the input information. (As done in many attention-based papers)

---

### Official Review · Reviewer_budh · 2023-11-01

**Soundness:** 2 fair
**Presentation:** 4 excellent
**Contribution:** 2 fair
**Rating:** 3
**Confidence:** 4

**Summary:**

This paper proposes a data-dependent long convolutional architecture DCLNet, which introduces data dependency though three modules: Layer-Wise Mapping, Rectify SideNet and SWEAP operator. The authors claim that "current long-range convolution have extensive parameter usage and limited in-context learning capabilities" and that DCLNet "surpasses the state-of-the-art baselines in processing lengthy sequences, even when trained in short sequences".

**Strengths:**

The paper is well-structured and easy to read.

**Weaknesses:**

I have multiple concerns in relationship to this work. In particular, the modules here introduced, namely Layer-Wise Mapping, Rectify SideNet and the SWEAP operator, seem to have been introduced by different previous papers under different names.

* First, the paper claims throughout the paper that “current long-range convolutions have problems with excessive parameter usage”. However, this is not the case for most long convolutional models. To give an example, Hyenas [1] rely on the parameterization of convolutional kernels of CKConv [2], which uses implicit neural representations to parameterize the kernel. This --which to the best of my understanding is at least analogous to the LayerWise mapping proposed here (see below for more discussions on this)--, also produces a kernel parameterization which is sublinear wrt the length of the sequence considered. Note that several other existing parameterizations of long conv models also exhibit the same behavior, e.g., S4 [3], H3 [4], S5 [5], etc.

  In fact, the layer-wise mapping proposed in this paper seems to be a less powerful replication of what has been done in CKConv [2] and Hyenas [1]. Specifically, both of these use an MLP to map coordinates to kernel values, which –just as here– allows for the generation of long conv kernels that can be made dynamic based on the input length. With less powerful I refer to the conclusion of the study in CKConv regarding high-frequencies. Note that both CKConv and Hyenas rely on “implicit neural representations” for the parameterization of the kernel. CKConv depicts that using normal MLPs leads to very bad parameterizations unable to depict high-frequencies.

* Secondly, the rectify sideNet is very similar to what is done in the Hyena operator of level 1. Specifically, the rectify sideNet can be understood as a combination of two input dependent mappings and a long conv. Given that typically level-2 Hyena’s are used, I do not see how this could be more powerful than Hyenas and, to the best of my understanding, the rectify sideNet seems to be a more constrained construction.

* Finally, the paper proposes to use a learnable exponentially decreasing mapping to mask on the generated kernel. However, the idea of using learnable masks to restrict the receptive field of the kernel has been proposed in FlexConv [6] with Gaussian masks, and exactly explored in Hyena and several other SSM papers in a learnable, exponentially decreasing form.

Altogether, I feel that the method proposed in this paper is a restricted form of Hyenas with multiple heads that have different nonlinearities. However, all these different relationships have not been properly described in the paper. I would like to note that it is not per se a problem that methods are related. However, if this is the case, I consider it important to discuss how the proposed method is different and why these simplifications make sense.

* Given the previous observations, I do not understand the reasoning behind the “partial dependency” and “fully dependency” in Table 1. Based on the discussion in the previous points, the only difference I see between these two classes is the use of a different nonlinearity in the gating operation.

* *Experimental results.* In addition, I am not sure that the experimental experiments support the claim of the paper in the abstract. Specifically, the paper states that: “Extensive experiments have demonstrated that DCL-Net surpasses the state-of-the-art in processing lengthy sequences, even when trained in short sequences.” However:

  First, in Table 2, this is not the case.

  Next, in Table 3, one would say that Hyenas and DLCNets are roughly on pair.

  On Table 4, this claim is also not supported.

  Finally, on Table 5, this is not compared to Hyena’s, which –based on my previous observations about the method– should be able to do this as well, I feel this comparison is incomplete.

**Questions:**

Aside from the previous observations I have a few other questions & notes:

* In Eq. 17, is this division stable?

* Between Eqs.17-18, I am not sure I understand how the influence of padding is removed and what is its effect.

* As explained before, CKConv and Hyenas use a different parameterization for the MLPs used to handle high-frequencies. Do you have any thoughts on the implications of modeling high-frequencies for the current paper?


# Conclusion

Based on my previous comments, this paper seems to present a simplification of Hyenas –which seems to be also supported by the experimental results–. Given my previous comments I cannot support acceptance of the paper in its current form. I want to emphasize that a simplification of Hyenas could be an interesting development. However, given the discussions that are still missing as well as the disparities between the claims and the results of the paper, I am against accepting the paper in the current form.

# References

[1] Poli, Michael, Stefano Massaroli, Eric Nguyen, Daniel Y. Fu, Tri Dao, Stephen Baccus, Yoshua Bengio, Stefano Ermon, and Christopher Ré. "Hyena hierarchy: Towards larger convolutional language models." arXiv preprint arXiv:2302.10866 (2023).

[2] Romero, David W., Anna Kuzina, Erik J. Bekkers, Jakub M. Tomczak, and Mark Hoogendoorn. "Ckconv: Continuous kernel convolution for sequential data." arXiv preprint arXiv:2102.02611 (2021).

[3] Gu, Albert, Karan Goel, and Christopher Ré. "Efficiently modeling long sequences with structured state spaces." arXiv preprint arXiv:2111.00396 (2021).

[4] Dao, Tri, Daniel Y. Fu, Khaled K. Saab, Armin W. Thomas, Atri Rudra, and Christopher Ré. "Hungry hungry hippos: Towards language modeling with state space models." arXiv preprint arXiv:2212.14052 (2022).

[5] Smith, Jimmy TH, Andrew Warrington, and Scott W. Linderman. "Simplified state space layers for sequence modeling." arXiv preprint arXiv:2208.04933 (2022).

[6] Romero, David W., Robert-Jan Bruintjes, Jakub M. Tomczak, Erik J. Bekkers, Mark Hoogendoorn, and Jan C. van Gemert. "Flexconv: Continuous kernel convolutions with differentiable kernel sizes." arXiv preprint arXiv:2110.08059 (2021).

---

### Meta-Review · Area_Chair_JEro · 2023-12-07

**Metareview:**

**Summary**
The authors propose a data-dependent long convolution model, namely DLCNet, that enables in-context learning for convolutional sequence learning. The model include 3 new operations: 1) Layer-wise mapping to keep minimal number of parameters, 2) Rectify SideNet to extract data-dependent kernel, and 3) SWEAP to filter noise out. The resulting model outperforms transformer-based models with similar size on multiple benchmarks.


**Strengths**
- Multiple reviewers find this paper well-written, although one reviewer thinks the presentation still can be improved.


**Weaknesses**
- Relationship to existing works is not clear. Similar operations have been proposed (pointed out by reviewer budh)
- Result wise, DLCNet seems not significantly better than other alternatives.

**Justification For Why Not Higher Score:**

Two reviewers are negative, and one reviewer is positive with a low confidence. The author did not respond during the discussion period. From the reviews, there are a lot of unaddressed questions, and the support from the positive review does not outweigh those issues. Therefore, my recommendation is reject.

**Justification For Why Not Lower Score:**

N/A

---

### Decision · Program_Chairs · 2024-01-16

Reject